# Structural Characterization of Rat Galectin-5, an N-Tailed Monomeric Proto-Type-like Galectin

**DOI:** 10.3390/biom11121854

**Published:** 2021-12-09

**Authors:** Federico M. Ruiz, Francisco J. Medrano, Anna-Kristin Ludwig, Herbert Kaltner, Nadezhda V. Shilova, Nicolai V. Bovin, Hans-Joachim Gabius, Antonio Romero

**Affiliations:** 1Department of Structural and Chemical Biology, CIB Margarita Salas, CSIC, Ramiro de Maeztu 9, 28040 Madrid, Spain; fruiz@cib.csic.es (F.M.R.); fjmedrano@cib.csic.es (F.J.M.); 2Physiological Chemistry, Department of Veterinary Sciences, Ludwig-Maximilians-University Munich, Lena-Christ-Str. 48, 82152 Planegg-Martinsried, Germany; Anna-Kristin.Ludwig@tiph.vetmed.uni-muenchen.de (A.-K.L.); kaltner@tiph.vetmed.uni-muenchen.de (H.K.); gabius@tiph.vetmed.uni-muenchen.de (H.-J.G.); 3Shemyakin & Ovchinnikov Institute of Bioorganic Chemistry, Russian Academy of Sciences, 16/10 Miklukho-Maklaya str., 117437 Moscow, Russia; pumatnv@gmail.com (N.V.S.); professorbovin@yandex.ru (N.V.B.)

**Keywords:** β-hairpin, β-sandwich, blood group B, lectin, sugar code

## Abstract

Galectins are multi-purpose effectors acting via interactions with distinct counterreceptors based on protein-glycan/protein recognition. These processes are emerging to involve several regions on the protein so that the availability of a detailed structural characterization of a full-length galectin is essential. We report here the first crystallographic information on the N-terminal extension of the carbohydrate recognition domain of rat galectin-5, which is precisely described as an N-tailed proto-type-like galectin. In the ligand-free protein, the three amino-acid stretch from Ser2 to Ser5 is revealed to form an extra β-strand (F0), and the residues from Thr6 to Asn12 are part of a loop protruding from strands S1 and F0. In the ligand-bound structure, amino acids Ser2–Tyr10 switch position and are aligned to the edge of the β-sandwich. Interestingly, the signal profile in our glycan array screening shows the sugar-binding site to preferentially accommodate the histo-blood-group B (type 2) tetrasaccharide and N-acetyllactosamine-based di- and oligomers. The crystal structures revealed the characteristically preformed structural organization around the central Trp77 of the CRD with involvement of the sequence signature’s amino acids in binding. Ligand binding was also characterized calorimetrically. The presented data shows that the N-terminal extension can adopt an ordered structure and shapes the hypothesis that a ligand-induced shift in the equilibrium between flexible and ordered conformers potentially acts as a molecular switch, enabling new contacts in this region.

## 1. Introduction

Storage of biological information involves more than nucleic acids and proteins. The ubiquity of occurrence, the enormous diversity already at the level of oligomers and the fine-tuned spatiotemporal regulation of the appearance of distinct structures are solid arguments for a fundamental functional meaning of the glycan part of cellular glycoconjugates [1,2,3,4,5,6]. Indeed, by molecular complementarity of oligosaccharides with a contact region in the carbohydrate recognition domains (CRDs) of sugar-binding proteins (lectins), glycan-encoded messages are ‘read’ and ‘translated’ into cellular effects [6,7,8]. Toward this end, triggering specific bioeffects, not only the selection of the binding partner(s), appears to matter. Furthermore, the lectin’s design, modularity and quaternary structure are first revealed in the case of the tetrameric leguminous lectin concanavalin A by a lower extent of crosslinking of certain cell surface receptors [9,10]. Fittingly, the context of presentation of the CRD shows a wide range of variability within lectin families. When considering the emerging multifunctionality of lectins, regions not involved in glycan binding can also affect their mode of action. Variability in design and the potential of regions beyond the glycan-binding site to be a physiologically relevant call for a detailed structural analysis within the lectin families in all their naturally occurring forms.

Focusing on the adhesion/growth-regulatory ga(lactose-binding) lectins, their common CRD is presented in three types of protein architecture, i.e., as (non)covalently associated homo/heterodimers (proto or tandem-repeat types) or as the chimera-type galectin-3 (Gal-3) with its N-terminal stalk attached to the CRD [11,12,13,14]. This highly dynamic, over 100-amino-acid-long sequence is composed of sections of known functionality; i.e., non-triple helical collagen-like repeats (for self-association) and an N-terminal peptide with two sites for serine phosphorylation (for intracellular compartmentalization) [12,15,16,17]. Two members of this lectin family are peculiar: galectin-related protein (GRP) and rat galectin-5 (rGal-5) present a short N-terminal extension (of up to 37 amino acids) of the canonical CRD of unknown function, which is clearly a challenge to study. Their special status as N-tailed proto-type-like proteins thus prompted to accomplish structural characterization of the full-length protein. In particular, it is of interest to define the structural features of the N-terminal extension, if adopted. Since respective attempts had so far been unsuccessful in the cases of human and chicken GRP, which had been crystallized as a truncated version [18,19], rGal-5 is the remaining target protein to try a full characterization of an N-tailed proto-type-like galectin.

This lectin was first purified from rat lung (denoted as RL-18) [20]. Sequencing of the cDNA from a rat reticulocyte library identified a strong homology (over 80%) to the C-terminal CRD of the tandem-repeat-type galectin-9 (Gal-9C), and monitoring among mammalian genomes disclosed its status as being uniquely present in rats [21,22,23,24]. The exon profile assumed that the rGal-5 gene originated from a species-specific gene duplication event followed by partial deletion to maintain the first exon coding for 13 amino acids and then the three exons of Gal-9C [12,24] (Figure 1). Notably, duplications and copy number variability of the galectin genes between species are not uncommon among mammals [25], rGal-5 being a specially processed species-specific form.

In solution, the current status of analysis describes rGal-5 as a monomer with a weak haemagglutinin activity [20,22,26]. Non-sialylated glycan termini are binding partners, especially when clustered, as is the case for N-acetyllactosamine (LacNAc) of the three complex-type N-glycans of the nonavalent pan-galectin-binding glycoprotein asialofetuin, and rGal-5 binding distinguishes late-stage apoptotic from secondary necrotic peripheral blood lymphocytes [27,28,29,30]. Selectivity in glycan binding is also implied in rGal-5′s involvement in the sorting processes during reticulocyte membrane remodeling by exosomal release [31].

In our study, we first report the binding specificity of rGal-5 through a glycan array, followed by further analysis of the ligand binding by isothermal titration calorimetry (ITC). The top places of a blood-group tetrasaccharide in the array test are due to its high affinity relative to LacNAc and is envisioned by evidence-based docking. Having succeeded in obtaining crystals of full-length rGal-5, structural information is then presented on the lectin in the absence and in the presence of lactose (Lac). The detected difference in the arrangement of the N-terminal tail in ligand-free and -loaded protein might suggest a molecular switch controlling contact formation in this area.

## 2. Materials and Methods

### 2.1. Protein Production and Purification

Recombinant production and purification by affinity chromatography followed by controls to ascertain purity were performed as previously described [26,32]). Afterwards, the rGal-5 and GRIFIN protein samples were extensively dialyzed at 277 K against 5 mM sodium phosphate buffer (pH 7.2), 0.2 M NaCl and 4 mM β-mercaptoethanol (PBS_β_). Finally, the galectin-containing solutions were concentrated using Amicon Ultra 10,000 MCWO centrifugal filter units (Millipore, Darmstadt, Germany), and then loaded into a Hi-Prep 16/60 Sephacryl S100 column (GE Healthcare, Freiburg, Germany) equilibrated with 20 mM Na-K phosphate buffer (pH 7.0), 150 mM NaCl and 4 mM β-mercaptoethanol. To obtain the cGRIFIN–tetrasaccharide complex, the purified protein sample was then incubated with the sugar at a 1:2 molar ratio in the same buffer for 10 min in ice. In the case of the rGal-5–lactose complex, the sample was purified by affinity chromatography using home-made lactosylated Sepharose 4B [33] immediately after the first dialysis. The protein bound to the resin was eluted with PBS_β_ and 200 mM lactose, concentrated and loaded into a Hi-Prep 16/60 Sephacryl S100 column (GE Healthcare, Freiburg, Germany) equilibrated with the previously described buffer supplemented with 5 mM lactose. All solutions of sugar-free and -loaded proteins were concentrated to a final concentration of 15 mg/mL for the screening of crystallization conditions. 

### 2.2. Glycan Array Measurements

The array consisted of 609 compounds covering glycans and polysaccharides printed onto commercial NHS-activated Slide H (Schott Nexterion, Jena, Germany), tested with 100 µg/mL biotinylated rGal-5 and involving Alexa555-labeled streptavidin (Thermo Fisher Scientific, Eugene, OR, USA) as second-step reagent for Innoscan 1100AL scanner-based (Innopsys, Carbonne, France) signal qualification, expressed in medium relative fluorescence units (RFU) and medium absolute deviation (MAD), as described previously [34,35]. When the fluorescence intensity exceeded the background value by a factor of five, the respective signal was considered to be significant.

### 2.3. Crystallization, Data Collection and Processing

Protein crystals were grown at 295 K by the vapor diffusion method. Specifically, rGal-5 and rGal-5–lactose crystals grew in 10% PEG 8000, 100 mM Tris pH 7.0 and 200 mM MgCl_2_. Crystals were soaked in this solution supplemented with 30% v/v glycerol as cryoprotectant. GRIFIN–tetrasaccharide crystals were grown in 15% w/v PEG 400 and 100 mM MES pH 6.5. The same solution supplemented with 25% w/v PEG 400 was used as cryoprotectant. All crystals were flash-cooled by immersion in liquid nitrogen. Diffraction data were collected at the beamlines BM14 of the ESRF Synchrotron (Grenoble, France) and BL13-XALOC of the ALBA Synchrotron (Cerdanyola del Valles, Barcelona, Spain). Crystallographic data were processed using XDS [36] and Aimless [37]. Details of the diffraction data are presented in Table 1.

### 2.4. Structure Determination and Refinement

The Molecular Replacement Method was used to solve the structures. A poly-Ala model based on the structure of Gal-9C (PDB entry 3NV1 [38]) was used to determine the structure of rGal-5. The PDB entry 5NMJ [32] was used as the search model to solve the cGRIFIN–tetrasaccharide structure. Structural refinements were carried out using Phenix [39]. Manual building, addition of water molecules and placement of ligands were done using Coot [40]. Details of the model refinements are given in Table 1. Protein–protein interactions, in particular those engaging the rGal-5 N-terminal residues, were analyzed using the PISA web server [41]. Figures for structural representation were drawn with the Pymol program [42].

### 2.5. Analytical Ultracentrifugation

Galectin-containing samples were diluted to a final protein concentration of 0.45 mg/mL in buffers for size-exclusion chromatography and pre-cleared by a centrifugation step at 16,000× *g*. Sedimentation velocity experiments were run at 293 K in an Optima KL-I analytical ultracentrifuge (Beckman Coulter, Krefeld, Germany) with an An50-Ti rotor and standard double-sector Epon-charcoal center pieces (1.2 cm optical path length). Measurements were performed at 48,000 rpm, registering successive entries every minute at 280 nm. Rayleigh interferometric detection was used to monitor the profile of the concentration gradient as a function of time and radial position, and the data were analyzed using SedFit software (Version 14.7).

### 2.6. SAXS Experiments

SAXS data were collected at the beamline BM29 (ESRF Synchrotron, Grenoble, France) using the BioSAXS robot and a Pilatus 1M detector (Dectris, Switzerland) with synchrotron radiation at a wavelength of λ = 1.000 Å and a sample-detector distance of 2.867 m [43]. Each measurement consisted of 10 frames of 1 s exposure each for a 100 μL sample flowing through a 1-mm-diameter capillary during X-ray exposure. Buffer scattering was measured immediately before each measurement of the corresponding protein sample at 277 K. The obtained scattering images were spherically averaged and the buffer scattering intensities subtracted using in-house software. Protein samples were prepared at concentrations of 4 mg/mL and 6 mg/mL in 20 mM Na-K phosphate buffer at pH 7.0 containing 150 mM NaCl and 5 mM lactose. Particle envelopes were generated ab initio using the program DAMMIF [44]. Multiple runs were performed to generate 20 independent model shapes that were combined and filtered to produce an averaged model using the DAMAVER software package [45].

### 2.7. ITC Measurements

The spacered B (type 2) tetrasaccharide was synthesized as described previously [46]. Titrations were monitored in a PEAQ-ITC calorimeter (Malvern, Westborough, MA, USA), using a galectin-containing solution of 250 µL in PBS (10 mM Na_2_HPO_4_, 2 mM KH_2_PO_4_, 137 mM NaCl and 3 mM KCl at pH 6.8) containing 10 mM β-mercaptoethanol and injections at 150 s intervals of 2 µL ligand-containing solution (up to adding 36.4 µL) at 25 °C (750 rpm), as described [33,47]. The protein concentration was 125 μM, and the ligand concentration in the syringe was 2.5 mM. In each titration, a fitted offset parameter was applied to account for potential background. Data processing was performed using the MiroCal PEAQ-ITC Analysis software.

## 3. Results

### 3.1. Glycan Array Data

rGal-5 is first tested to determine its binding profile to chip-presented substances, mostly glycans up to the molecular mass of bacterial polysaccharides. By using an array platform with 609 compounds, the spacered histo-blood-group B (type 2) tetrasaccharide, LacNAc-based dimers and the xenoantigen with α1,3-linked galactose added to a LacNAc core were found to be frontrunners in terms of signal intensity, together with several bacterial polysaccharides (Figure 2; for a complete listing of compounds and signal intensities, please see Appendix A). rGal-5, in contrast to GRP, which has lost the ability to bind β-galactosides [19], thus presents a profile with typical selectivity among this class of glycans. To report the contact pattern between rGal-5 and the selected carbohydrate ligands, we then carried out systematic screening to find conditions for crystallization. In these experimental series, we used the full-length protein to obtain structural information on the N-terminal tail.

### 3.2. Overall Crystallographic Structure of Full-Length rGal-5

Ligand-free full-length rGal-5 crystallizes in the monoclinic P2_1_ space group and diffracts to a resolution of 1.7 Å. An estimation of the crystal solvent content suggested the presence of six galectin molecules in the asymmetric unit (Figure 3A). The rGal-5–lactose complex crystals belong to space group P22_1_2_1_, with only two molecules present in the asymmetric unit (Figure 4A), and diffract to a 1.9 Å resolution. 

The overall fold in ligand-free (Figure 3B) and -loaded rGal-5 (Figure 4A) is composed of two antiparallel β-sheets (F1 to F5 and S1 to S6 strands) that form the characteristic β-sandwich structure. A short 3_10_ helix is placed between strands F5 and S2. Beyond analyzing the architecture of the contact site for glycans (see below), these crystals offered the opportunity to examine whether the N-terminal extension presents well-ordered elements or high flexibility.

### 3.3. Structure of Ligand-Free rGal-5

In the ligand-free structure, six rGal-5 molecules are arranged in the asymmetric unit, as shown in Figure 3A. The core of the different protein units can readily be superimposed onto each other, as revealed by the low average root mean square deviation (RMSD) value among them of 0.26 Å for all Cα atoms. Differences are attributed mainly to the N-terminus, and the six protein monomers can be divided into two groups (Figure 3A), based on the experimental electron density. In the first group (chains A–C), the 11 residues at the N-terminus are not visible in the electron density map. Their likely extended and flexible structure in solution can indeed be derived from our SAXS data: the ab initio model of rGal-5 calculated on this basis exhibits, expectably, a globular shape. Most interestingly, though, a cylindrical extension on its top was seen in the model. When placing the CRD within the spherical region, the extended N-terminal section matches the geometry of the cylindrical part of the SAXS model (see below).

When inspecting the second group (chains D–F), the electron density for this peptide stretch was clearly observed: the three amino acids from Ser2 to Ser5 form an extra β-strand, named F0, running in antiparallel direction to the C-terminal F1 strand. Residues Thr6 to Asn12 are in a loop, placed in parallel to the axis of the β-sandwich and protruding more than 10 Å from the S1 and F0 strands (Figure 3B).

The conformation within this loop is stabilized by interactions with symmetry-related molecules: hydrogen bonds between Pro9–His142, Tyr10–Glu103 and Asn12–Gly52 as well as a water bridge between Asn12 and Asn74 (Figure 3C).

### 3.4. Structure of Ligand-Loaded rGal-5

The two CRDs present in the asymmetric unit, which have bound lactose (Lac), exhibit very similar features to ligand-free rGal-5 (Figure 4A), with an RMSD value of only 0.3 Å for all Cα atoms. One of the two monomers in the asymmetric unit exhibits strong electron density for the first 10 residues so that their structure could be modelled. Intriguingly, these residues run parallel to the edge of the β-sandwich (Figure 4A) instead of forming the F0 strand and the protuberant loop observed in the ligand-free state. Intramolecular (hydrogen bonds between Ser2 and Ser3 with Ser39) and intermolecular contacts with symmetry-related molecules (hydrogen bond between Thr6 and Asp40) stabilize this special spatial arrangement (Figure 4B). 

The carbohydrate-binding site in the concave face of the β-sheet is constituted by β-strands S4 to S6. The amino acids of the signature sequence, i.e., His57, Asn59, Arg61, Asn70, Glu80 and Arg82, directly interact with lactose through hydrogen bonding interactions. Additionally, Arg43, Gln45 and Glu64 form water-mediated hydrogen bonds with the ligand. As commonly found in galectin–lactose complexes, the indole ring of Trp77 stacks to the β-face of the pyranose ring of galactose (Figure 4C). In the absence of lactose, these residues form contacts with water (or glycerol molecules under conditions used for crystallization) molecules. Only minor rearrangements are observed for residues Arg43 and Glu64, which interact through water molecules with lactose (Figure 4D). Moving beyond defining the contact pattern, the thermodynamics of the ligand binding was analyzed by ITC.

### 3.5. ITC Measurements

rGal-5 interact with Lac with a dissociation constant of 136 ± 16 μM, which is lowered in the case of LacNAc to 30.5 ± 1.9 μM and 5.5 ± 0.6 μM for the blood-group B tetrasaccharide (Table 2 and Appendix A). This stepwise affinity enhancement can be explained by the increased number of contacts that these ligands make with additional amino acids. In this case, we used its complex with an avian galectin cGRIFIN (see below) shown in Figure 5 and the respective model building to obtain the relevant information for the new additional contacts, as described below, between the tetrasaccharide and rGal5.

### 3.6. Structure of Ligand-Loaded cGRIFIN


Our attempts to crystallize rGal-5 bound to the blood-group B tetrasaccharide were unsuccessful. Thus, we decided to test chicken GRIFIN (cGRIFIN) as a model for the binding of this compound. This very stable protein has been previously crystallized in several conditions [32]. We were able to obtain crystals of cGRIFIN in the presence of the blood-group B tetrasaccharide. These crystals diffracted up to a resolution of 1.14 Å (Table 1). This high-resolution data allowed us to build the sugar structure in the electron density in both carbohydrate-binding sites of the dimer. A comparison of the lactose-bound (PDB 5NLE) and the tetrasaccharide-bound cGRIFIN structures shows the absence of any significant structural change between these two structures, the RMSD value being 0.382 Å for all Cα atoms. The GalB moiety fully superposes with the galactose moiety of lactose, forming H-bonds with His46, Asn48, Arg50, Asn59 and Glu69. On the other hand, the GlcNacB moiety is rotated in the tetrasaccharide compared to the glucose moiety of lactose. Despite this change in the conformation, the H-bond with Glu69 is conserved. The acetamido group is exposed to the solvent as it is the FucA moiety. The GalA moiety establishes two additional H-bonds, one of them linking the 6′-hydroxyl group with the NE atom of Trp66. The second one extends the binding site beyond the S4 strand, linking the 2′-hydroxyl group with Glu32 (Figure 5A).

The superposition of the lactose-bound rGal-5 structure with the blood-group B tetrasaccharide-bound cGRIFIN gave an RMSD of 0.69 Å for all Cα atoms, showing the similarity of both complexes. This similarity allows us to analyze the interactions that could be established between this ligand and rGal-5 (Figure 5B). The GalB and GlcNacB moieties of the tetrasaccharide could stablish the same interactions as those observed for lactose, including the one with Arg82. The GalA moiety is properly placed to interact with the NE atom of Trp66 and with the side chain of Gln45, a residue from the S3 strand. In addition, Arg43 faces the FucA ring and could interact with this moiety, expanding the ligand-protein surface of contact (Figure 5C). This last residue belongs to the loop connecting the S3 and F2 strands, the region interacting with the N-terminal residues in the ligand bound rGal-5 structure.

### 3.7. Oligomerization State of Full-Length rGal-5

Despite the disparity in packing inside the asymmetric unit of rGal-5 crystals without and with a ligand, intramolecular interactions, computed using the PISA Web server [41], did not appear to be sufficient to promote oligomerization, in both cases.

We experimentally confirmed by analytical ultracentrifugation and small-angle X-ray scattering (SAXS) in solution the absence of any oligomerization. The range of protein concentrations from 0.45 mg/mL up to 6 mg/mL was covered to trace any tendency to form oligomers at high non-physiological concentrations. In sedimentation velocity experiments in the absence or presence of 0.1 M lactose, rGal-5 (at 0.45 mg/mL) appeared as a single peak with a sedimentation coefficient of 1.8 ± 0.1 S (s20, *w* value of 1.9 ± 0.1 S, after correcting for the effect of solvent density and viscosity) (Appendix A). This result is fully in line with a globular protein of a mass of 13.8 kDa, as calculated from its amino acid sequence. Small-angle X-ray scattering (SAXS) data for the rGal-5–lactose complex at the concentrations of 4 mg/mL and 6 mg/mL yielded a particle distribution that is also attributed to a molecular mass of approximately 13 kDa (Figure 6). These data sets further substantiate that rGal-5 is a monomer in solution (under these conditions), as it is in the obtained crystals.

## 4. Discussion

Gal-5 is an N-tailed proto-type-like galectin present exclusively in rats and exists in solution as a monomer. To fill the gap of the structural characterization of this particular protein and obtain information of a full-length N-tailed proto-type-like galectin, the three-dimensional structures of the *apo* form and in complex with lactose were determined by X-ray crystallography. The protein crystallizes as a monomer in the absence and in the presence of ligand. In solution, analytical ultracentrifugation and SAXS experiments extended the available evidence for the lack of any intermolecular association.

Within the ligand-binding site, the side chains of the residues of the signature sequence for sugar recognition do not adopt their position by ligand binding. In the ligand-free structure, a network of water molecules or the presence of a single glycerol molecule takes the place of the core of a cognate glycan in the preformed contact site. The validity of the concept for such an intimate preorganization that can also accommodate a compound with a sugar-like constellation of hydroxyls, such as glycerol, has been thoroughly documented for the CRD of human Gal-3 [48,49,50], also described for murine Gal-4′s N-terminal CRD [51]. In order to define cognate compounds, glycan array testing revealed preferential affinity of rGal5 for the histo-blood-group B determinant and its fucose-less trisaccharide as well as LacNAc-based tetrasaccharides among the tested set of mammalian glycans. The calorimetric titrations reflect the affinity gain for LacNAc and the blood-group B epitope relative to lactose (Table 2 and Appendix A).

Reflecting its proposed origin, rGal-5 shares structural features with Gal-9C (respective data available for the human protein) to a great extent. The availability of individual crystallographic information for human (h) Gal-9N and Gal-9C [52,53,54,55] made it possible to superimpose the rGal-5 structure to both Gal-9 CRDs. The calculated RMSD value is smaller for hGal-9C (0.37 Å) than for hGal-9N (0.56 Å). rGal-5 and hGal-9C could be overlaid almost perfectly, with loops occupying similar positions around the ligand-binding site (Figure 7A). As observed in the hGal-9N structure, a short β-strand is formed in the ligand-free rGal-5. This extension of the β-sandwich involves residues Pro9 and Tyr10 (Figure 7B). The N-terminal CRD of the human tandem-repeat-type Gal-9 thus mimics rGal-5′s tendency for gaining some order in the N-terminal extension.

Structural differences between rGal-5 and hGal-9C relative to the N-terminal CRD of hGal-9 were found in the loop regions by insertion or deletion of residues (Figure 7A). These differences can be linked to shifts in specificities in glycan-binding between both CRDs in hGal-9, such as a reduction in affinity towards the LacNAc oligomers for the C-terminal with respect to the N-terminal CRD [38,56,57]. Loops connecting the F2-S3 and S3-S4 strands have additional residues in hGal-9N, covering the S2-S4 β-strands and forming a highly favorable binding site for LacNAc and its oligomers (polyLacNAc repeats) in N- and O-glycans and keratan sulphate, interacting with both CRDs [56,57,58]. Binding of these ligands involves a hydrogen bond with Asn137 in the hGal-9N [38]. The corresponding residues in hGal-9C and rGal5 are Gly313 and Ala135, respectively, hindering this interaction. In addition, His223 (in hGal-9C) and Gln45 (in rGal5) occupy the equivalent position of Ala46 in hGal-9N, causing a steric impediment for ligand accommodation within this region of the S3 β-strand (Figure 7B). This residue was linked to the specificity of hGal-9N for polyLacNAc repeats, the Forssman pentasaccharide and the histo-blood-group A hexasaccharide [52,53]. The loops connecting the S4-S5 and the S5-S6 strands are shorter in rGal5 and hGal-9C than in hGal-9N. These loops form the entrance for the ligand-binding site, their similar shape letting rGal-5 and the C-terminal CRD of hGal-9 share affinity.

The distinctive characteristic of rGal-5 (and GRP) is the N-terminal extension to the canonical CRD. Since the galectin CRD can interact with binding partners beyond the site for accommodating lactose and can engage in two types of contact at the same time (e.g., Gal-3 binding glycan and the chemokine CXCL12 [59]), changes between flexible and ordered arrangements of a tail may establish a molecular switch to let a protein ligand dock or not onto this region. For example, the considerably longer tail in Gal-3 has been shown by ESI MS and NMR [60,61] to backfold. This move results in blocking the access to a region of the S-face of the CRD. The systematic design of the Gal-3 variants with truncated versions attached to the CRD facilitated the possibility of generating a new double-stranded antiparallel β-sheet at the F-face [62,63]. In that case, the obtained information indicated the potential for forming an ordered structural element in the distal section that may have a bearing on the presentation of the Ser acceptor for phosphorylation [62]. Identification of respective counterreceptors for rGal-5 with a contact at this site will be required to support such an idea, giving further work a clear direction. 

Interestingly, the monomeric C-type lectin RegIIIγ (HIP/PAP in mice) express a flexible N-terminal extension, which is a prosegment maintaining the protein in a biologically inactive state and is proteolytically removed to let the CRD become antibacterial via peptidoglycan binding [64]. Equally important, functional assays with an engineered variant to establish a protein pair (such as lectin with and without the N-terminal tail) can be informative for rGal-5 and GRP to trace the physiological significance for the extension, such as for the N-tailed proto-type-like galectins. The detection of the relevance of the isomer state of the Pro4 or Pro5 peptide bond in two galectins, chicken galectin-1B and human galectin-7, in the quaternary structure, illustrates the apparent fine-tuning of galectin activity by in-built molecular switches [65,66]. More work on the N-terminal tail is encouraged as a result of this.

## Figures and Tables

**Figure 1 biomolecules-11-01854-f001:**
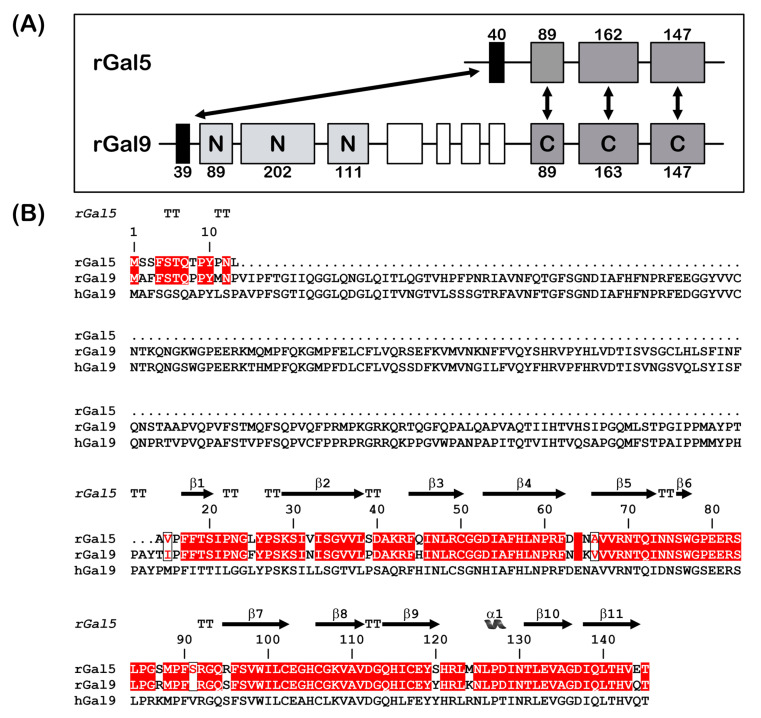
(**A**) Gene structures for rGal-5 and -9. The size of the exons (boxes) is indicated, and introns are drawn as lines (not drawn to scale). The N- and C-terminal CRDs of rGal-9 are labelled and given in different grey levels. Homologous exons are indicated (double arrows). (**B**) Sequence alignment of rGal-5 with rat and human Gal-9. Strictly conserved residues (red) and similar residues (boxed red letters) between rat proteins are shown. The upper lane represents the secondary structure elements of rGal-5 (α represents α-helices, β represents β-sheets and TT represents β-turns).

**Figure 2 biomolecules-11-01854-f002:**
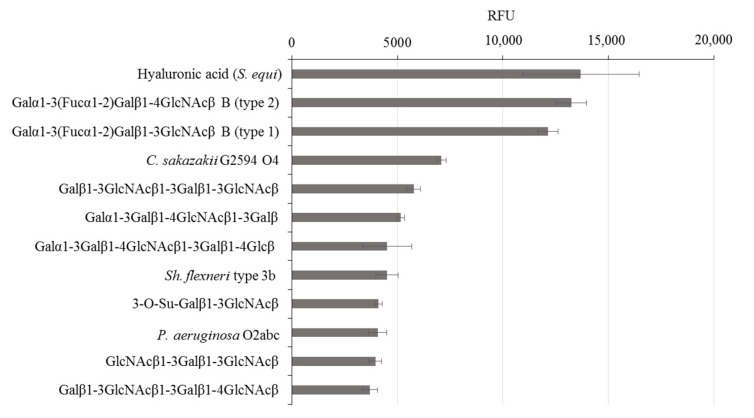
Top-12 glycans in the composition of the glycan array that exhibit binding with rGal-5.

**Figure 3 biomolecules-11-01854-f003:**
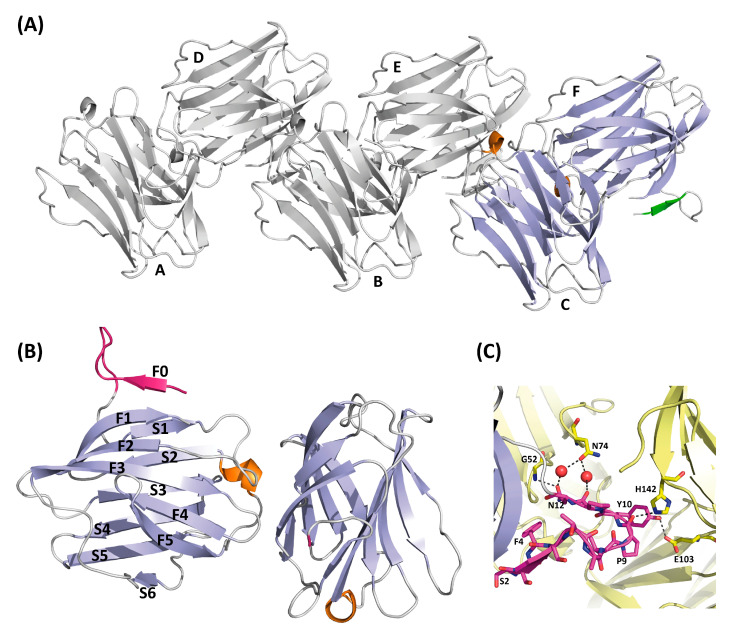
Structure of ligand-free rGal-5. (**A**) Ribbon diagram of the asymmetric unit of the crystals containing six molecules(A–F) of rGal-5 (helix in orange, β-strands in blue and the N-terminal extension in green). (**B**) Strands (labelled F0 to F5 on one side and S1 to S6 on the other) forming the characteristic β-sheets are labelled (helix in orange, β-strands in green and the N-terminal extension in magenta). An extra strand is found, named F0, placed immediately in front of a loop protruding from the CRD. (**C**) Intermolecular interactions stabilize the extended loop conformation. These interactions involve residues Gly52 (G52), Pro9 (P9), His142 (H142), Tyr10 (Y10), Glu103 (E103), Asn12 (N12), Gly52 (G52), Asn74 (N74). Symmetry-related molecules are shown in yellow.

**Figure 4 biomolecules-11-01854-f004:**
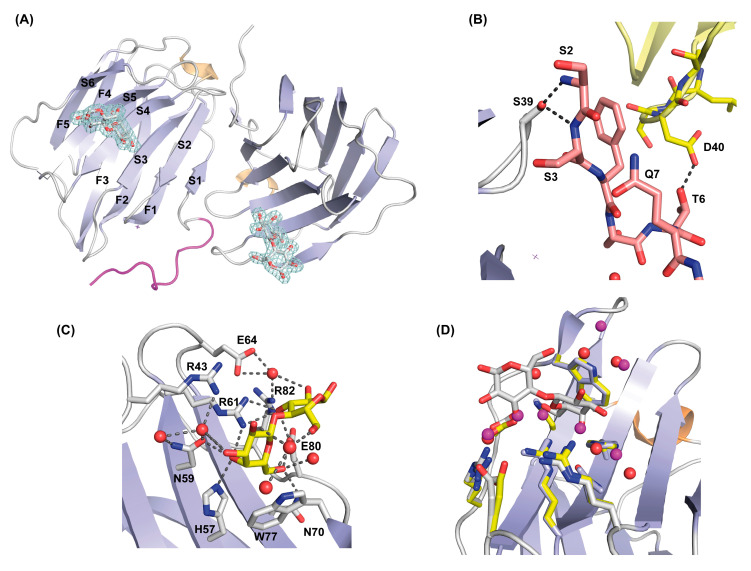
Structure of the rGal-5–lactose complex. (**A**) Overall architecture of the asymmetric unit of the rGal-5–lactose complex, with two CRDs; in one of them, the N-terminal residues (in magenta) adopt an extended geometry interacting with the edge of the β-sandwich. Strands that form the characteristic β-sheets are labelled (S1 to S6 on one strand, F1 to F5 on the other), and lactose molecules represented in sticks showing the 1.9 Å resolution 2F_o_-F_c_ electron density map (in blue) contoured at 1.0 σ. (**B**) Inter- and intra-molecular interactions that stabilize the extended conformation of the N-terminal residues. These interactions involve residues Ser2 (S2), Ser3 (S3), Ser39 (S39), Thr6 (T6) and Asp40 (D40). Symmetry-related molecules are represented in different colors. (**C**) Close-up view of the ligand-binding site of rGal-5 to show interactions between the protein and lactose. Key protein residues [His57 (H57), Asn59 (N59), Arg61 (R61), Asn70 (N70), Glu80 (E80) Arg82 (R82), Arg43 (R43), Glu64 (E64) and Trp77 (W77)] and lactose are represented in stick mode and water molecules as red spheres. (**D**) Superposition of the ligand-binding sites of rGal-5 (yellow) and the rGal-5–lactose complex (grey). Side-chain positions of residues at this site are not affected by ligand binding, indicating a preformed geometry. In the ligand-free structure, the position of the residues is kept by interactions with water or glycerol molecules. Water molecules from this last structure are shown in purple for clarity.

**Figure 5 biomolecules-11-01854-f005:**
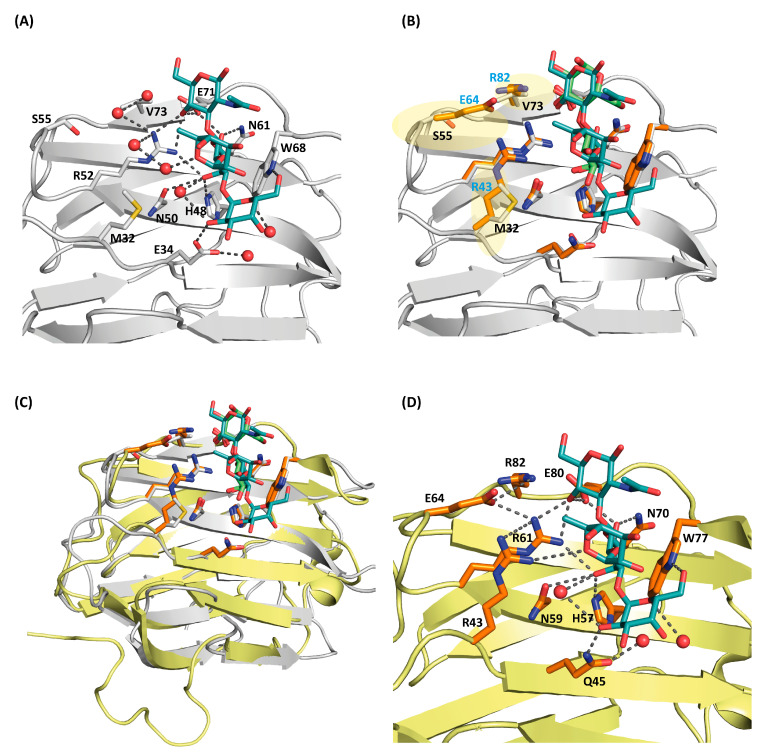
(**A**) Close-up view of the carbohydrate-binding site (CBD) of cGRIFIN showing the interactions between the histo-blood-group B tetrasaccharide and the active site residues [Met32 (M32), Glu34 (E34), His48 (H48), Asn50 (N50), Arg52 (R52), Ser55 (S55), Asn61 (N61), Trp68 (W68), Glu71 (E71) and Val73 (V73)]. (**B**) Structural comparison between cGRIFIN bound to the blood-group B tetrasaccharide complex (carbons in white) with the rGal-5/lactose complex (lactose in green, carbons of the interacting residues in orange). While residues such as His57, Asn59, Arg61, Asn70, Glu80 and Arg82 occupy almost the same position than residues in the former structure, the presence of charged residues (Arg53, Glu64 and Arg82) (highlighted in semitransparent yellow color) may lead to direct or water-mediated interactions with the ligand. (**C**) Superposition of the cGRIFIN/tetrasaccharide structure (grey; the histo-blood-group B tetrasaccharide in cyan) with the rGal-5–lactose complex (yellow; lactose in green) highlighting the differences in the active site residues between cGRIFIN and rGal-5 [Met32 (M32) to Arg43 (R43), Ser55 (S55) to Glu64 (E64), and Val73 (V73) to Arg82 (R82)]. (**D**) The rGal-5/histo-blood-group B tetrasaccharide modeled by superposition with the cGRIFIN/tetrasaccharide structure. The active site residues involved in bindig of the ligand are: Arg43 (R43), Gln45 (Q45), His57 (H57), Asn59 (N59), Arg61 (R61), Glu64 (E64), Asn70 (N70), Trp77 (W77), Glu80 (E80) and Arg82 (R82). Residues Arg43, Glu64 and Arg82 might explain the affinity of rGal-5 for the histo-blood-group B (type 2) tetrasaccharide.

**Figure 6 biomolecules-11-01854-f006:**
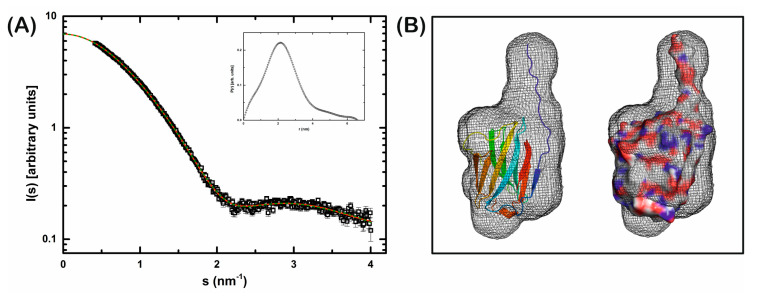
Small-angle X-ray scattering (SAXS) experiment supports a monomeric quaternary structure of rGal-5 in solution. (**A**) SAXS scattering profile of rGal-5. Black squares represent the experimental data, the red line the theoretical fitting obtained with the program GNOM. Inset: pair distances distribution function. (**B**) Ab initio SAXS model generated with the program DAMMIN (grey mesh). The crystallographic structure of the rGal-5 CRD domain with additional N-terminal residues modelled in an extended conformation is shown inside the envelope.

**Figure 7 biomolecules-11-01854-f007:**
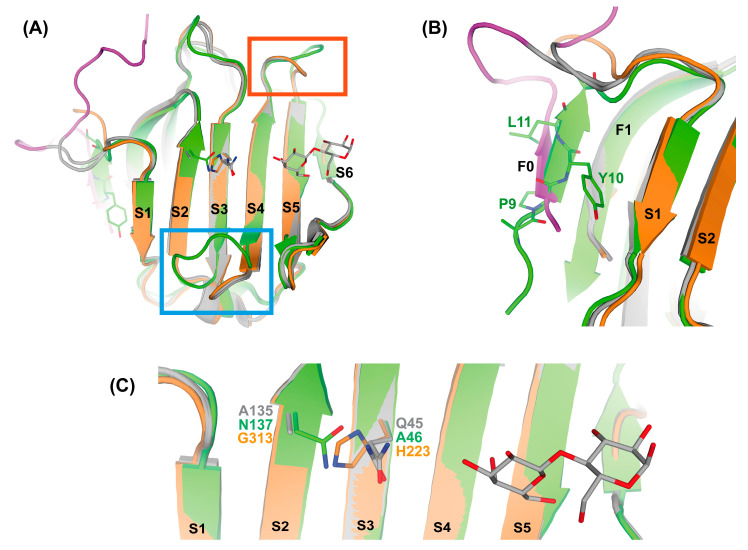
(**A**) Structural comparison of rGal-5 (grey) with the N-terminal (green) and the C-terminal (orange) CRD domains of hGal-9 showing the closer relationship between the first and the last structure. Differences in loops connecting strands in the S-face (labelled S1 to S6) are highlighted with colored squares. (**B**) Close view of N-terminal residues showing the formation at the F0 strand in rGal5, as observed in the structure of the N-terminal domain for hGal-9. (**C**) Specificity for LacNAc motif repeats in hGal-9N involves a hydrogen bond with Asn137 (N137) in the hGal-9N, while the corresponding residues in hGal-9C and rGal5 are Gly313 (G313) and Ala135 (A135), respectively. Furthermore, His223 (H223, in hGal-9C) and Gln45 (Q45, in rGal5) occupy the equivalent position of Ala46 (A46) in hGal-9N, causing steric impediment.

**Table 1 biomolecules-11-01854-t001:** Data collection and refinement statistics.

	rGal-5	rGal-5 + Lactose	cGRIFIN + Tetrasaccharide
Data collection			
Space group	P 2_1_	P 22_1_2_1_	P 22_1_2_1_
Cell dimensions			
a, b, c (Å)	66.1, 68.1, 95.4	39.0, 65.8, 112.7	39.1, 70.6, 87.7
A, β, γ (°)	β = 91.8		
Resolution(Å) ^a^	33.03–1.70	39.02–1.90	43.85–1.13
	(1.76–1.70)	(1.97–1.90)	(1.17–1.13)
Total reflections	402258 (17733)	46992 (4574)	181470 (17698
Unique reflections	92950 (9247)	23584 (2322)	90857 (8884)
R_merge_	0.057 (0.457)	0.060 (0.252)	0.015 (0.305)
R_meas_	0.077 (0.623)	0.085(0.947)	0.021 (0.431)
CC 1/2	0.99 (0.84)	0.99 (0.84)	1.00 (0.90)
Completeness (%)	99.75 (99.21)	99.78 (99.74)	99.13 (97.73)
<I/σ(I)>	14.01 (2.80)	7.60 (2.74)	22.0 (2.0)
Wilson B-factor	13.96	15.69	9.99
Multiplicity	4.1 (3.7)	2.0 (2.0)	2.0 (2.0)
Refinement			
R_work_	0.17 (0.23)	0.17 (0.22)	0.15 (0.42)
R_free_	0.21 (0.28)	0.23 (0.29)	0.17 (0.43)
Nº atoms (non-hydrogens)	7547	2533	2926
Protein	6659	2212	2350
Ligands	42	47	184
Water	846	274	482
Protein residues	834	279	276
Average B factor (Å^2^)	18.06	16.67	14.52
Protein atoms	16.60	15.61	12.39
Ligands	31.46	15.19	14.45
Water	28.88	25.47	24.93
R.m.s. deviations			
Bond lengths (Å)	0.006	0.007	0.007
Bond angles (°)	0.81	0.88	1.05
Ramachandran statistics			
Favoured (%)	98	98	95.59
Outliers (%)	0.5	1.2	0.37
Clashscore	5.7	1.79	3.56
PDB code	5JP5	5JPG	7P8H

^a^ Values in parentheses are for the highest-resolution shell.

**Table 2 biomolecules-11-01854-t002:** ITC data for ligand binding to recombinant rGal-5 (at 25 °C).

Ligand	K_d_ (μM)	Stoichiometry	ΔG^0^_obs_ (kcal/mol)	ΔH^0^_obs_ (kcal/mol)	−TΔS^0^_obs_ (kcal/mol)

Lactose	121 ± 5	0.95 ± 0.05	−5.35	−4.99 ± 0.08	−0.36
	151 ± 5	0.99 ± 0.06	−5.22	−5.03 ± 0.41	−0.18

LacNAc	28.6 ± 2.0	0.92 ± 0.01	−6.1	−10.0 ± 0.2	3.87
	32.3 ± 7.1	0.94 ± 0.20	−6.0	−10.5 ± 2.6	4.43

Tetrasacc haride	6.1 ± 0.2	0.94 ± 0.01	−7.11	−5.60 ± 0.03	−1.51
	5.0 ± 0.2	0.98 ± 0.01	−7.11	−5.74 ± 0.03	−1.37


## Data Availability

The structures have been deposited at the Protein Data Bank and are available with the identifications listed in Table 1.

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
