# Peer review of "Structural Characterization of Rat Galectin-5, an N-Tailed Monomeric Proto-Type-like Galectin"

_biomolecules, 2021, doi:10.3390/biom11121854_

Round 1

Reviewer 1 Report

This article reports the structure characterization of a full-length galectin, both with and without ligand. These crystal structures revealed structural changes due to ligand binding and structural features of carbohydrate recognition domains. It was shown that this structural change became a molecular switch which controlled the ligand binding.

This article is well written and I recommend to publish after minor revision.

  1. Are there any differences in the lattice structures of ligand-binding and ligand-free structure ?
  2. The ligands in Figure 4C and in Figure5 are difficult to see, so you should change the color of the ligands.

Author Response

Referee 1.

  1. Are there any differences in the lattice structures of ligand-binding and ligand-free structure?

 - A typographical error was detected in Table 1. The unit cell constants for the rGal5-lactose complex are 39.0 65.8 112.7 instead of 39.0 65.9 and 12.7

- Lattice structures of both ligand-binding and ligand-free rGal5 are different. In the ligand-free structure the monoclinic P21 form contain 6 molecules in the asymmetric unit. However, the lactose complex is orthorhombic with 2 molecules in the asymmetric unit. Both dimers are not superimposable between both crystal forms.

  1. The ligands in Figure 4C and Figure 5 are difficult to see, so you should change the color of the ligands.

- Figures 4C and 5 have been modified, and colors of the ligands have been changed accordingly, so the new ones are now more reader friendly.

Reviewer 2 Report

The article entitled: Structural characterization of rat galectin-5, an N-tailed monomeric proto-type-like galectin - by Ruiz FM et al.

reports the crystallographic information on the N-terminal extension of the carbohydrate recognition domain of rat galectin-5, described as an N-tailed prototype-like galectin.

This work provides precise structural information on the sugar binding site of the molecule and a preferential binding to the histo-blood group B (type 2) tetrasaccharide and N-acetyllactosamine-based di- and oligomers has been demonstrated.

Interestingly experimental data show that the N-terminal extension can adopt an ordered structure and shapes the hypothesis that a ligand-induced shift in the equilibrium between flexible and ordered conformers can potentially modulate new molecular interaction at this site.

The work presented is very technical but is biologically relevant for better uderstand the structural and mechanistic bases of galectins’ multifunctionality. Regions not involved in glycan binding can also affect the function.

The binding specificity of rGal-5 through a glycan array (609 compounds), followed by further analysis of the ligand binding by isothermal titration calorimetry 91 (ITC) is also reported.

After a series of specific and well-performed structural and functional experiments the authors conclude that the distinctive characteristic of rGal-5 (and GRP) is the N-terminal extension to the canonical CRD.

Since the galectin CRD can interact with binding partners beyond the site for accommodating lactose and can engage in two types of contact at the same time, changes between flexible and ordered arrangements of a tail may establish a molecular switch to let a protein ligand dock or not onto this region.

Concluding the work is technically complex for the general readers but could be relevant for scientists working in the field of glycobiology in particular on galectins. For this reason I suggest the publication on biomolecules Journal

Author Response

We would like to thank you for your contribution

Reviewer 3 Report

The manuscript, titled “Structural characterization of rat galectin-5, an N-tailed monomeric proto-type-like galectin” by Ruiz et al., describes the results of the structural analysis of galectin-5 and discusses its property including its sugar-binding ability. This work is well described, is sound science, and could be important to researchers in this field. However, the points below should be addressed before publication.

Major comment: Page 9, Section 3.6. and Figure 5

I understood that the crystal of rGal-5 with blood-group B tetrasaccharide was not obtained. However, to discuss the binding between rGal5 and Blood-group B tetrasaccharide using cGRIFIN as a model, it is considered to be helpful to add a figure showing the expected binding between rGal5 and blood-group B tetrasaccharide in Figure 5.

For example, one of the following could be considered

(1) Docking simulation of rGal5 and Blood-group B tetrasaccharide

(2) Superimposition of rGal5 with cGRIFIN and glycans

Minor comment #1: Page 6, Figure 2

The fifth and sixth rows from the top are both “Galb1-3GlcNAcb1-3Galb1-3GlcNacb”. Is it a typographical error?

Minor Comment #2: Page 8, Line275

Instead of “an avian galectin”, “an avian galectin cGRIFIN” would be easier to understand.

Author Response

Referee 3.

Major comment. Figure 5. ....one of the following should be considered:
(1) Docking simulation of rGal5 and Blood-group B tetrasaccharide
(2) Superimposition of rGal5 with cGRIFIN and glycans

- The Figure 5 has been modified as advised. A new panel (C) has been included following the recommendation of the reviewer. The structures of rGal5/lactose and the cGRIFIN/Blood-group B tetrasaccharide have been superimposed.

Minor comment #1: The fifth and sixth rows from the top in Figure 2 are both “Galb1-3GlcNAcb1-3Galb1-3GlcNacb“. Is it a typographical error?

- The repeated structure was removed, so we incorporated one more glycan up to 12.

Minor comment #2: Instead of “an avian galectin“ , a“an avian galectin cGRIFIN“ would be easier to understand

- Agree too